**Subject Category:**
Biology (whole organism)

ecology/evolution

Calanoida, Cyclopoida, colorimetric method, predation, ultraviolet radiation, freshwater

**Author for correspondence:**
Marcus Lee
e-mail: marcus.lee@biol.lu.se

# Low-latitude zooplankton pigmentation plasticity in response to multiple threats

Marcus Lee[1], Huan Zhang[1,2], Yongcui Sha[1], Alexander Hegg[1], Gustaf Ekelund Ugge[1], Jerker Vinterstare[1], Martin Škerlep[1], Varpu Pärssinen[1], Simon David Herzog[1], Caroline Björnerås[1], Raphael Gollnisch[1], Emma Johansson[1], Nan Hu[1], P. Anders Nilsson[1,3], Kaj Hulthén[1,4], Karin Rengefors[1], R. Brian Langerhans[4], Christer Brönmark[1] and Lars-Anders Hansson[1]

[1]Department of Biology, Aquatic Ecology, Lund University, Lund, Sweden
[2]Chinese Academy of Sciences, Institute of Hydrobiology, Wuhan, People's Republic of China
[3]Department of Environmental and Life Sciences, Karlstad University, Karlstad, Sweden
[4]Department of Biological Sciences and W.M. Keck Center for Behavioral Biology, North Carolina State University, Raleigh, NC, USA

  ML, 0000-0002-3320-3010; YS, 0000-0002-8558-0125; AH, 0000-0003-4794-8352; JV, 0000-0001-6998-4632; MŠ, 0000-0001-5812-2383; CB, 0000-0002-3506-0367; RG, 0000-0001-6177-8877; PAN, 0000-0002-3541-9835; RBL, 0000-0001-6864-2163; L-AH, 0000-0002-3035-1317

Crustacean copepods in high-latitude lakes frequently alter their pigmentation facultatively to defend themselves against prevailing threats, such as solar ultraviolet radiation (UVR) and visually oriented predators. Strong seasonality in those environments promotes phenotypic plasticity. To date, no one has investigated whether low-latitude copepods, experiencing continuous stress from UVR and predation threats, exhibit similar inducible defences. We here investigated the pigmentation levels of Bahamian 'blue hole' copepods, addressing this deficit. Examining several populations varying in predation risk, we found the lowest levels of pigmentation in the population experiencing the highest predation pressure. In a laboratory experiment, we found that, in contrast with our predictions, copepods from these relatively constant environments did show some changes in pigmentation subsequent to the removal of UVR; however, exposure to water from different predation regimes

induced minor and idiosyncratic pigmentation change. Our findings suggest that low-latitude zooplankton in inland environments may exhibit reduced, but non-zero, levels of phenotypic plasticity compared with their high-latitude counterparts.

## 1. Introduction

All organisms are perpetually exposed to information conveying both threats and opportunities [1,2], and each individual organism must act upon this information to maximize its opportunities while simultaneously minimizing the risk from threats. In prey organisms, a common strategy to reduce risk in the face of increasing threat from predators is to induce defence traits, such as specific behaviours, morphologies and chemicals that reduce an individual's vulnerability to predation [3,4]. Inducible defences are favoured when there is temporal variability in predation pressure and when prey have reliable means of evaluating predation risk [3], whereas temporally homogeneous environments should select for canalized phenotypes [5], i.e. prey defence traits that are constitutive and locally adapted to the prevailing predation regime. Furthermore, inducible defence traits should incur costs to the individual's fitness which prevents them from becoming constitutive defences, which are expressed even in the absence of the threat [6]. Despite their ephemeral nature, plastic defensive traits have profound effects for both direct and indirect interactions with other organisms, which make them both ecologically and evolutionarily influential [7].

In aquatic systems, zooplankton comprise a long-standing and valuable model for investigating inducible defences [8–12]. They are amenable to laboratory experimentation and occupy an integral position in aquatic food webs, filling the role of primary consumers as well as being an indispensable prey source for most larval and many adult fish [13]. Zooplankton frequently induce modifications in morphology, physiology and behaviour to gain protection from predators [9,12,14,15]. For example, rotifers such as *Keratella* spp. can induce spine elongation when exposed to the predaceous rotifer *Asplanchna* sp. [16] or decrease spine length when threatened with fish predators [17]. Cladocerans, particularly in the genus *Daphnia*, boast a plethora of inducible defences: they form helmets, invest in longer tail spines, induce diapause and alter swimming behaviours, all in response to predator cues [6,9,18,19]. These predator cues, or kairomones, are detected through chemoreception which informs the zooplankton of general rather than acute predation risks [20].

Threats do not only arise from the risk of predation, however. Solar ultraviolet radiation (UVR) is another well-documented stressor for zooplankton, eliciting multiple forms of inducible protection [8,14]. Copepods, a common and important group of zooplankton, have demonstrated inducible defences in response to UVR exposure [2,21]. A common strategy is to accumulate photoprotective compounds such as melanin, mycosporine-like amino acids or carotenoids [22]. In environments where UVR is a substantial threat, such as clear lakes which allow UVR to penetrate deeper [23], copepods have been shown to contain large quantities of carotenoids, which confer protection from UVR through the neutralization of free radicals [24,25]. In their free form or as bound lipids, carotenoids appear red or yellow [22]. This pigmentation can make individuals more conspicuous targets for visually hunting predators such as fish [26]. Therefore, it should be predicted that copepods in environments with fish predators will have lower levels of pigments than nearby populations without visually hunting predators. Numerous studies—primarily at high latitudes or in high-elevation lakes—have demonstrated that this trade-off exists and that copepods can rapidly adjust pigmentation levels in response to changes in predation cues or UVR [2,14,21,26,27]. Due to seasonal changes in these environments, there is a substantial variation in both UVR and predation levels across the year, and this variation is a key feature for the promotion of phenotypic plasticity and inducible defences [3,28].

Multiple ecological and environmental differences between temperate and low-latitude systems can influence the plasticity of pigmentation. Fish reproductive periods, for example, are more constrained towards the poles as fish typically reproduce once annually, whereas fish in the subtropics are fractional spawners resulting in a less variable predation regime analogous to the climatic variability hypothesis [29,30]. Similarly, UVR in high-latitude environments is particularly stressful during summer, but the threat is completely absent during winter months; in the subtropics, however, the threat of UVR is still variable but never absent (figure 1). To our knowledge, no studies yet have investigated the plasticity of pigmentation at lower latitudes.

Bahamian 'blue holes', which are water-filled vertical caves with a freshwater layer floating atop marine ground water [31], represent a unique opportunity to investigate zooplankton pigmentation and the role of phenotypic plasticity in the subtropics. They represent isolated, temporally stable environments that are very simple with regard to the trophic webs as they only contain a small number of species [31].

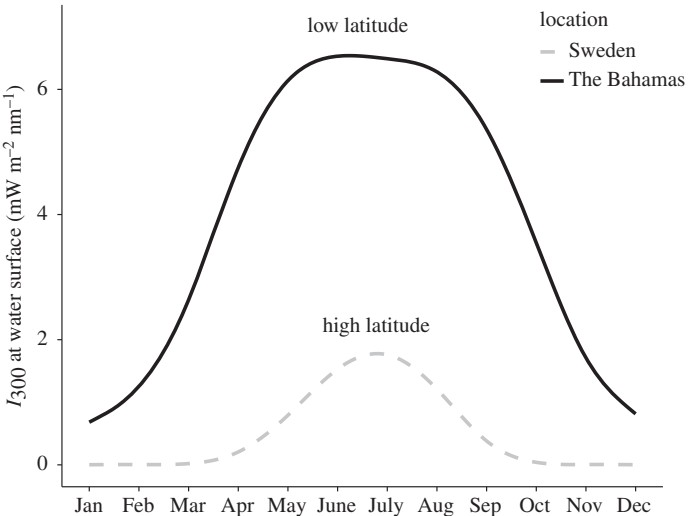

**Figure 1.** Estimation of surface irradiance of UV-B (300 nm) at a high-latitude location (Lund, Sweden; 55.71°N, 13.20°E) and a low-latitude location near the focal study sites (Nassau, The Bahamas; 25.05°N, 77.40°W) across the year during 2017. Data computed from the 'FASTRT' model V2.3 (https://fastrt.nilu.no) and computed as the first day of each month at noon with cloudless conditions.

Importantly, as these blue holes were formed thousands of years ago, the zooplankton in these systems have evolved with different apex predators: some blue holes have both piscivorous and zooplanktivorous fish, others have only zooplanktivorous fish and a few lack vertebrate predators altogether [31]. This range of stable predator regimes, coupled with the intense while still variable, year-round UVR, allows us to make explicit predictions regarding the level of pigmentation expected in copepods inhabiting environments with conflicting threats.

The objective of the present study was to identify whether low-latitude copepods show similar phenotypic patterns to those from high latitudes when exposed to UVR and predation pressure. We hypothesized that the high and more constant UVR exposure across the subtropics requires year-round protection in all blue holes and, hence, by comparing blue holes that differ in fish assemblage, we could test the hypothesis that copepods from environments with greater threats of predation from visually hunting predators would have lower levels of photoprotective pigmentation. Specifically, we predicted that zooplankton in environments with no predators should have the highest level of pigmentation, those with only zooplanktivorous fish would have the lowest, and due to the reduced yet not absent predation pressure, those with both zooplanktivorous and piscivorous fish would have an intermediate level. Furthermore, in a laboratory experiment, we tested the prediction that low-latitude copepods, unlike copepods from higher latitudes, will not exhibit phenotypic plasticity in pigmentation due to the low temporal variation in predation intensity and the continual presence of UVR over the year, with any phenotypic differences between populations instead representing constitutive defences.

# 2. Material and methods

## 2.1. Field sampling

Copepods were collected from three blue holes on Andros Island, The Bahamas, during March 2018. Blue holes, although sharing most characteristics, do vary in many features such as surface area, freshwater depth and turbidity. Therefore, we selected these blue holes *a priori* primarily based upon the presence/absence of zooplanktivorous and piscivorous fish [31] while also considering the geographical proximity to one another and the similarity of the UVR threat among blue holes (see electronic supplementary material). Turtle Blue Hole (24°46′21.72″N, 77°51′5.472″W) has no fish, hereafter 'no-predation'; Cousteau's Blue Hole (24°46′33.6″N, 77°54′57.6″W) harbours a population of a piscivorous fish species (bigmouth sleeper, *Gobiomorus dormitor*) and a relatively low density of a zooplanktivorous fish (Bahamas mosquitofish, *Gambusia hubbsi*), hereafter referred to as 'low-predation'; and Rainbow Blue Hole (24°47′6″N, 77°51′36″W) has no piscivorous fish and a high density of *G. hubbsi*, hereafter 'high-predation'. We estimated the daytime predation risk of small aquatic prey in each blue hole by quantifying the average number of bites towards possible prey by fish per minute per cubic metre

(see electronic supplementary material). These estimates confirmed our expectation that Rainbow Blue Hole experienced the highest predation risk, followed by Cousteau's Blue Hole, and then Turtle Blue Hole which had no fish predators (electronic supplementary material, table S1). To collect the copepods, we sampled the entire freshwater layer of each blue hole by lowering a 100 µm net, with a mouth diameter of 30 cm, and gently retrieving it. We collected a sample of concentrated zooplankton for immediate pigmentation analysis and another sample for a laboratory experiment (see below). We found that no- and high-predation systems contained only calanoid copepods and the low-predation system contained only cyclopoid copepods. As we were interested in general patterns and both types of copepods exhibit plasticity in pigmentation at higher latitudes [2], we included both in our laboratory experiment.

## 2.2. Experimental design

To test whether pigmentation in copepods at lower latitudes is a phenotypically plastic trait that responds to changes in UVR and predation risk, we performed a laboratory experiment. To assess the influence of predation risk, we employed a $3 \times 3$ factorial experimental design (3 populations × 3 treatments) with five replicates each. Population represented the initial predation regime (no-, low- and high-predation risk) of the population and treatments represented differences in perceived predation risk (chemical predator cues). We devised our treatments using water from the three blue holes filtered through a 50 µm mesh to remove other large zooplankton but keep both phytoplankton and the chemical cues from any potential predators. We collected experimental animals using the methodology above, and all were collected on the same day and brought to the laboratory where there was a 12 : 12 light : dark photoperiod with no exposure to UVR. This absence of UVR allowed us to explicitly test whether the pigmentation of these copepods is plastic in response to UVR as well as perceived predation risk. For each regime (no-, low- and high-predation risk, respectively), we filled fifteen 300 ml containers with water collected in the field and again filtered through a 50 µm mesh. We then took the zooplankton samples from the field and divided each population into each container: five replicates of the no-predation risk system water, five replicates of the low-predation risk system water and five replicates of the high-predation risk system water (45 total containers, 5 replicates per population × treatment combination). This was achieved by gently mixing the field samples before taking a 200 ml subsample, thereby coarsely standardizing the number of animals, filtering the water away using the 50 µm mesh and carefully introducing the zooplankton on the mesh to the treatment container.

The laboratory experiment was run for 10 days, as it has been shown that carotenoid content can adjust to changed risk levels after only 4 days [21]. To maximize the predator cues, while simultaneously limiting mechanical disturbance to the copepods, 100 ml of water from each container was exchanged every other day. This was achieved by slowly filtering through a 50 µm mesh (to ensure that experimental zooplankton were not lost) and replaced with 100 ml of filtered treatment water collected that same day from the respective blue hole. This method also supplemented each container with fresh phytoplankton and micro-zooplankton for both the herbivorous and omnivorous copepods. At the end of the experimental period (10 days), each container was randomly selected, gently mixed and the water filtered through a 50 µm mesh to collect the copepods. Only live adult copepods were photographed for the quantification of pigmentation using the methodology described below.

## 2.3. Quantification of pigmentation

To quantify pigmentation, live copepods were gently transferred to an individual drop of glycerol on a glass slide using forceps. To avoid any damage to the cephalosome (our area of interest, as to minimize the risk of measuring the green gut material), animals were manipulated by their antennae. We then took a digital photo of each copepod at 200× magnification using a Dino-Lite Edge X 200x (USB3) microscope (AnMo Electronics Corporation, Taiwan) and the associated DinoXcope software. All copepods were manipulated into the same position when taking photos, and light conditions were standardized by taking the photos in a darkened room with only the focal light on the subject. To assess pigmentation levels, the photos were subjected to a profile conversion in Adobe Photoshop CC 2017, following Brüsin et al. [15]. Here, the colour profile is changed from RGB to Lab Colour, which is based upon the standardized, device independent colour space, CIE (Commission International de l'Eclairage) $L^*a^*b^*$. Still in Photoshop, the 'Quick Selection tool' was used to select the cephalosome, and the mean 'redness' ($a^*$) and 'yellowness' ($b^*$) in that given selection were obtained from the in-built histograms. The values for those colour channels range from 0, which appears as true green ($a^*$) or true blue ($b^*$) to the human eye, to 255, which appears true red ($a^*$) or true yellow ($b^*$). Together, these two attributes of colour have successfully been applied as proxies for carotenoid-based coloration [32,33].

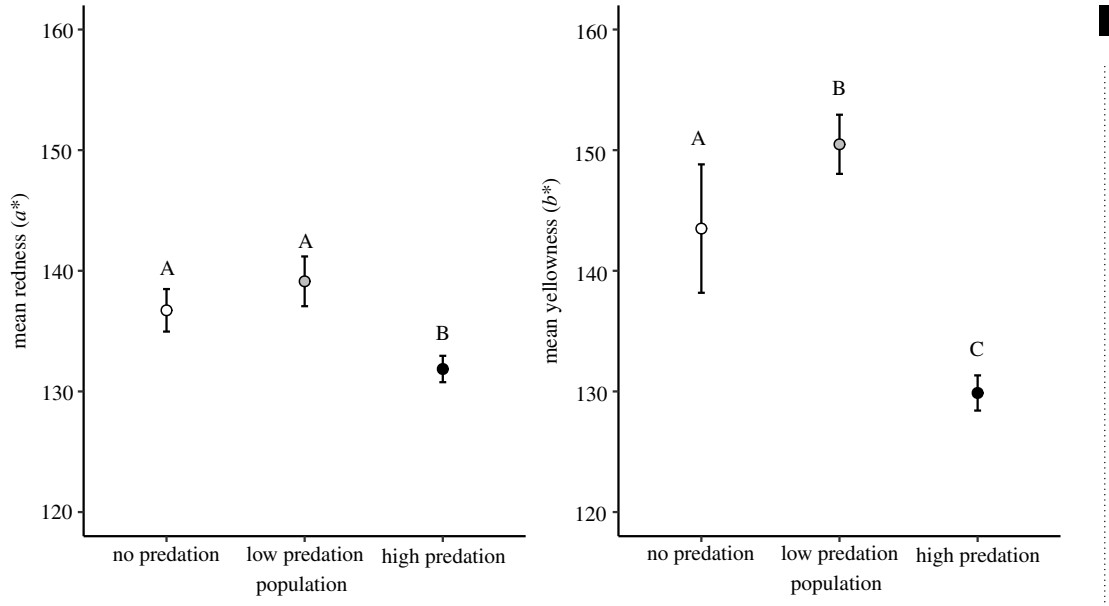

**Figure 2.** Mean copepod pigmentation levels ($a^*$ and $b^*$) $\pm$ 2 s.e. among three Bahamian blue holes with different food chain lengths and, therefore, predation risk ranging from no fish (no-predation risk, white bars), both zooplanktivorous and piscivorous fish (low-predation risk, grey bars) and only zooplanktivorous fish (high-predation risk, black bars) ($n = 6$ per lake). Horizontal lines represent the mean with the box denoting the 25th and 75th percentiles and the whiskers representing the 5th and 95th percentiles. Letters denote significant difference between populations ($p < 0.05$).

## 2.4. Statistical analysis

All analyses were performed using R v. 3.4.3 [34]. To test for differences among the blue hole populations, we conducted two separate ANOVAs using redness and yellowness values, respectively, as dependent variables and employed the *post hoc* Tukey's test to examine differences between populations if the main effect was statistically significant ($p < 0.05$). To examine effects of population and water source (i.e. predation cue) in the laboratory experiment, we calculated the change ($\Delta$) in pigmentation by subtracting the colour values of the zooplankton exposed to our treatments at the end of the experiment from the average pre-experiment value of the population from which they originated. We then performed separate linear mixed models using $\Delta a^*$ and $\Delta b^*$ values as dependent variables with the package 'lme4' [35]. Treatment, population and their interaction served as fixed effects, while replicate ID was entered as a random effect as five individuals were taken from each container, yielding a sample size of 25 for each population × treatment combination.

## 3. Results

### 3.1. Population differentiation

Field-collected copepods exhibited clear differences in pigmentation between populations ($a^*$: $F_{2,15} = 19.24$, $p < 0.0001$; $b^*$: $F_{2,15} = 36.17$, $p < 0.001$; figure 2). In accordance with our predictions, environments with only zooplanktivorous fish (high predation) had the lowest levels of carotenoid pigmentation. This high-predation regime exhibited lower levels of both redness and yellowness compared with those from the fishless environment (Tukey's tests: $p = 0.003$ and $p < 0.001$, respectively) or compared to those from the low-predation system (Tukey's tests: both $p < 0.001$). Unexpectedly, the cyclopoid copepods from the low-predation system had similarly high levels of redness as the copepods from the no-predation system (Tukey's test, $p = 0.143$) and even greater levels of yellowness (Tukey's test: $p = 0.032$).

### 3.2. Laboratory experiment

As UVR is continually present over the year at low latitudes, we expected little-to-no effects of UVR removal across populations in the common garden environment. However, contrary to our expectations, we found

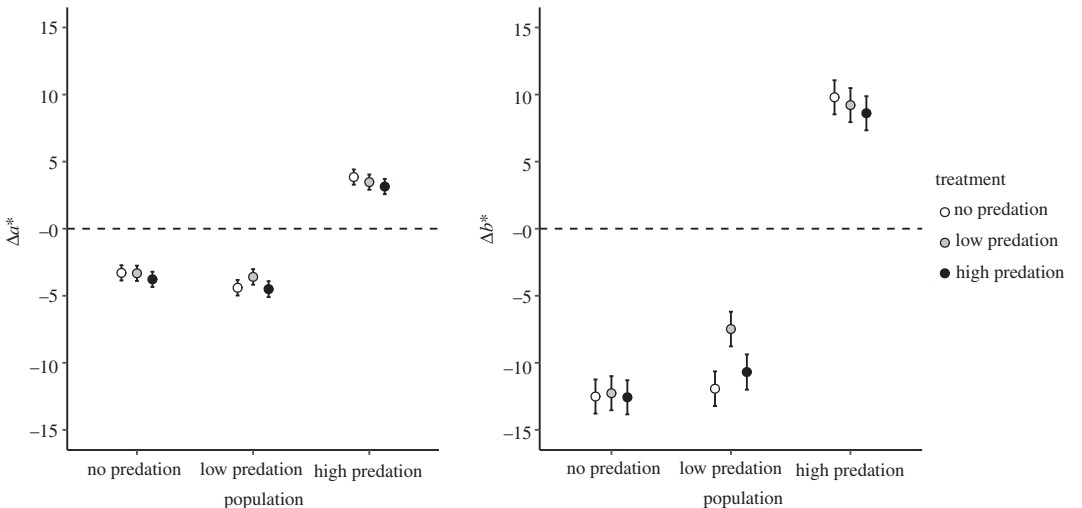

**Figure 3.** Change in mean pigmentation levels ($\Delta a^*$ = redness and $\Delta b^*$ = yellowness) of copepods ($n = 25$ per treatment $\times$ population) due to the removal of UVR and exposure to predatory cues in the laboratory experiment compared with the mean pigmentation of samples taken directly from the respective blue hole (hatched line). The values depicted here represent the least square means and the $\pm 2$ standard error taken from the independent mixed models, accounting for the effect of the replicates. Increases in pigmentation are denoted by positive values and losses are denoted by negative values. All changed significantly from the baseline data ($p < 0.05$).

**Table 1.** Results from linear mixed-model analyses of changes in pigmentation, separately examined using redness ($a^*$) and yellowness ($b^*$) colour variables, during the laboratory experiment. Significant results are italicized.

| factor | colour channel | $F$ statistic | d.f. | $p$-value |
|---|---|---|---|---|
| treatment | $a^*$ | 3.14 | 2,35.3 | 0.056 |
| | $b^*$ | 4.60 | 2,32.5 | *0.017* |
| population | $a^*$ | 654.21 | 2,35.3 | *<0.001* |
| | $b^*$ | 1040.40 | 2,32.5 | *<0.001* |
| treatment $\times$ population | $a^*$ | 1.12 | 4,35.3 | 0.362 |
| | $b^*$ | 4.57 | 4,32.5 | *0.005* |

that the removal of the UVR threat resulted in clear changes in the pigmentation levels, albeit in different directions in the different populations (table 1 and figure 3; electronic supplementary material, figure S1). Regardless of the predation risk treatment, the no- and low-predation risk populations lost pigmentation (in both redness and yellowness) in the absence of UVR, whereas the high-predation risk population increased pigmentation. Second, we expected that due to the lack of temporal variation in predation pressure, we would not find plasticity in response to differing predation risk chemical cues. Contrary to our expectations, the introduction of water from alternative blue holes with contrasting predation regimes had minor effects upon copepod pigmentation (table 1 and figure 3). The effects of the treatment on $\Delta a^*$ were weak and marginally non-significant, although the treatment effects on $\Delta b^*$ were more pronounced (table 1). Lastly, we investigated whether there was an interaction between population and the treatment effects. We found no significant evidence for an interaction between treatment and population in $\Delta a^*$ but we did for $\Delta b^*$ (table 1). This result appears to be driven by the zooplankton in the low-predation system, which reduced their yellowness more when exposed to foreign water compared to water from their own blue hole (figure 3).

## 4. Discussion

There is a pronounced bias in plankton plasticity research towards high latitudes and high elevations where environmental conditions are highly seasonal and variable [11,21,24,26]. As such, it is unknown

whether zooplankton from low latitudes (i.e. subtropics and tropics) exhibit plasticity in phenotypic responses to divergent threats such as predation and UVR, as found in other systems. We postulated that copepods in the subtropics would follow similar among-population phenotypic patterns in pigmentation to those found at higher latitudes with respect to the prevailing threat, yet they would represent constitutive defences due to invariable predation risk and consistently present solar UVR, rather than the plastic responses so characteristic of copepods in higher latitudes [2]. Specifically, we hypothesized that copepods from environments with no visually hunting predators would have higher levels of photoprotective pigmentation than those exposed to fish. We also hypothesized that when removed from UVR and exposed to water from different predation regimes, they would not adaptively alter pigmentation according to the new threat regime.

Using the natural Bahamian blue hole system, we have been able to demonstrate that low-latitude freshwater copepods do in fact display similar among-population pigmentation patterns compared to their high-elevation and high-latitude counterparts [2,14]. That is to say, as predicted, calanoid copepods in an environment with zooplanktivorous fish (high predation) had less redness and less yellowness than those from the environment without visually hunting predators (no-predation). This finding is well aligned with the available literature concerned with the trade-offs in pigmentation protection and the presence of visually hunting predators, which suggests that the accumulation of carotenoid compounds appears to be restricted to environments from which fish predators are absent [2,21]. Despite the difference in species composition, we confirmed our prediction that the environment with both piscivorous and zooplanktivorous fish (low-predation) would have a higher level of pigmentation than the high-predation system due to reduced predation intensity through the predator effects on *Gambusia hubbsi* by *Gobiomorus dormitor*. However, our prediction was not met regarding the difference between no- and low-predation risk systems, which may be partly due to taxonomic differences of the copepod assemblage among blue holes. The low-predation risk system was dominated by cyclopoid copepods, whereas the other systems were dominated by calanoid copepods. Even if it is known that both cyclopoid and calanoid copepods increase pigmentation when exposed to high UVR levels, it is possible that taxonomic differences may affect pigmentation levels among these systems as it is known that even copepods from within the same family show differences in the ability to sequester pigmentation [36]. Therefore, despite the fact that cyclopoid and calanoid copepods have been shown to exhibit similar levels and seasonal variations in pigmentation at higher latitudes [2], we cannot state whether the level of predation or the independent evolution of calanoids and cyclopoids have led to the different levels of pigmentation observed here. Increasing the number of investigated lakes with different trophic levels from one per treatment would provide far clearer information on the relative importance of predation in pigmentation.

Having determined that there were phenotypic differences between differing species and predation regimes, the next step was to investigate whether the populations exhibited pigmentation plasticity in response to reduced UVR or changes in perceived predation risk. At higher latitudes, changes in UVR and predation pressure have been repeatedly demonstrated to induce pigmentation changes in copepods [2,21,22,24,26]. Copepods have also been shown to behaviourally respond to UVR and can therefore clearly sense the presence of UV wavelengths [37]. In these higher latitude environments, there are periods when UVR is absent and it is then beneficial to have low levels of carotenoids that are costly to maintain. As such, we posited that pigmentation would not be a plastic trait in the lower latitudes due to the constant presence of UVR year-round even though there is variability within the year (figure 1). Despite our predictions, we found that pigmentation is a plastic trait in the low-latitude blue hole systems.

The removal of UVR caused copepods from all populations to change their pigmentation. Copepods from the no- and low-predation systems reduced pigmentation, similar to other UVR removal experiments [27], whereas those from the high-predation system increased pigmentation levels (figure 3). We assume that copepods in high-predation systems are exposed to a high and constant threat of predation, and an adaptive behaviour in this environment towards both UVR and predation could be to avoid the surface waters during the day via diel migration, as opposed to the reduction in damage through carotenoid usage. Copepods are capable of detecting depth through the combination of hydrographic and optical features [38]. Our experiment mimicked surface waters and prevented the diel vertical migration behaviour that is possible in the blue hole system; therefore, the copepods may have increased pigmentation to protect against the surface UVR regime they have evolved to expect irrespective of predation, leading to the observed pattern. Furthermore, in the absence of hydrodynamic disturbances caused by fish predators, copepods from the high-predation population may have increased pigmentation in response to a perceived reduction in predation risk irrespective of the water-cue treatment [39].

In contrast with UVR conditions, changes in exposure to non-familiar predator cues had either no or only a minor effect on copepod pigmentation. The change in redness in copepods treated with water from the other populations was small (marginally non-significant treatment effects and no interaction effect), corresponding to our initial prediction that there would be no plasticity. The water-cue treatment had, however, a significant effect on yellowness, yet this appears to be driven by the low-predation population dominated by cyclopoid copepods. Water cues only altered yellowness in the low-predation system, where cyclopoid copepods exhibited reduced yellowness when exposed to water from either a no-predation or high-predation environment. This neither follows adaptive predictions nor our initial hypothesis. Carotenoids in copepods must be sequestered from the food source and can appear as either red or yellow. Consequently, the low-predation risk system may have had a phytoplankton composition with a higher proportion of 'yellow' carotenoid species, such as lutein that is found in Prasinophyceae and Chlorophyceae [40], than in the other systems. If this was true, however, it would be expected that the calanoid copepods from no- and high-predation systems would also have higher levels of $\Delta b^*$ when exposed to low-predation treatment water. But, as this was not the case, it appears that both pigmentation sequestering and the plasticity of this trait in this system vary among taxonomic groups as found in other studies [15,36].

Our initial prediction was that the temporal consistency in predation pressure and the intensity of UVR of the low-latitude blue holes would lead to canalized phenotypes. However, the clear and pronounced changes in pigmentation when UVR threat was removed indicate that these copepods have a pigment defence against UVR that is phenotypically plastic, not entirely constitutive. This may be due to the annual variation in UVR also present in the subtropics (figure 1). Despite UVR being a constant threat, i.e. never zero irradiance, like the winter months in the higher latitudes, the variation present could still be sufficient to promote plasticity. Furthermore, changes in cloudiness may create variable UVR conditions at a smaller temporal scale. As for plasticity in response to predation, we found only minor and idiosyncratic responses, fitting our expectations based on the temporal consistency in predation pressure in this system. It is possible that factors that promote and maintain plasticity, other than temporal variation in predation threat, may explain the minor degree of plasticity we observed here. For example, infrequent migration of copepods between blue holes that differ in the predation regime may contribute to the evolution and maintenance of plasticity [41]. We believe that our findings add to the mounting evidence that copepod plasticity is not as highly constrained at lower latitudes as earlier thought [42]. Specifically, the pigmentation response to UVR across all copepods is less constrained than previously thought, possibly due to the underestimation of the variation in the environmental cue. Further studies should address the mechanisms maintaining plasticity in low-latitude environments.

We conclude that zooplankton from different populations have differing pigmentation based upon the prevailing threat combination. Calanoid copepod populations in Bahamas blue holes exhibited pigmentation patterns matching predictions based on predation threat, similar to patterns previously observed at higher latitudes. High-latitude zooplankton also show adaptive plasticity in pigmentation in response to predator cues, while we here found that low-latitude calanoid copepods showed little evidence of plasticity and cyclopoid copepods exhibited only minor plasticity inconsistent with adaptive hypotheses.

Ethics. To collect zooplankton, no ethical permissions were required at that time. We were also not required to complete an ethical assessment prior to conducting our research as copepods are not a protected or legislated group. All animals were euthanized as humanely as possible at the end of the experiment and destroyed to prevent secondary toxicity. Fieldwork was conducted under R.B.L.'s permission from The Bahamas government.

Data accessibility. Raw data files, images used for the data collection and the R code used to analyse that data are available online in the Dryad Data Repository at: https://doi.org/10.5061/dryad.bd4486s [43]. Variables used to *a priori* select focal blue holes and a further figure describing the variation in the experiment are available within the electronic supplementary material.

Authors' contributions. M.L., H.Z., Y.S., A.H. and L.-A.H. conceived and led the study. M.L., R.B.L. and P.A.N. conducted the analyses. M.L. wrote the first version with substantial contributions from R.B.L. and C.B. All authors aided in fieldwork and provided valuable guidance both during the research process and the revision of the manuscript.

Competing interests. Authors declare no competing interest and confirm that the manuscript has not been submitted elsewhere for publication.

Funding. Financial support was provided by the Helge Ax:son Johnsons foundation, the Royal Physiographic Society and the Swedish Research Council (grant no. 2016-03552).

Acknowledgements. Deep thanks go to Samuel Hylander for stimulating conversation regarding copepod pigmentation. We also thank Eleanor O'Brien for initial statistical support and Rebecca Moss for proofreading. We thank The Bahamas government for permission to conduct the work and Wilfred Johnson for support in the field.

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
