## [Reviewer comments · Royal Society Open Science]

Review History

RSOS-190321.R0 (Original submission)

Review form: Reviewer 1

Is the manuscript scientifically sound in its present form?

Yes

Are the interpretations and conclusions justified by the results?

Yes

Is the language acceptable?

Yes

Is it clear how to access all supporting data?

Yes

Reports © 2019 The Reviewers; Decision Letters © 2019 The Reviewers and Editors; Responses © 2019 The Reviewers, Editors and Authors. Published by the Royal Society under the terms of the Creative Commons Attribution License <http://creativecommons.org/licenses/by/4.0/>, which permits unrestricted use, provided the original author and source are credited

Do you have any ethical concerns with this paper?

No

Have you any concerns about statistical analyses in this paper?

I do not feel qualified to assess the statistics

Recommendation?

Accept with minor revision (please list in comments)

Comments to the Author(s)

Dear authors,

It was a pleasure to reading the manuscript on pigmentation plasticity in copepods from low-latitude habitats you have submitted to the Royal Society Open Access Journal. The Blue Holes obviously provide an ideal opportunity to study community responses to environmental conditions, in this case UV exposure and predation pressure. It is, however, a little bit unfortunate that the community composition was so drastically different among the three holes, i.e. cyclopoids in no and high predation environment and calanoids in the intermediate system. This topic is addressed in the discussion (lines 240 ff) but I would suggest to removing this from results (maybe mention in the methods section), as you clearly state that such comparison does not yield much information.

In the detailed comments you will also find a question as to how many copepods were examined. I gather that it was always five individuals from each container, yielding 25 measurements for each group (see also Fig. 3 legend). It would be nice if you could state this more clearly in the methods section.

When I started working on the manuscript, I downloaded a zip file with the pictures of the copepods. Now, that I am finalizing my review, I tried to find the zip file on the web site again but I have not been successful, maybe because I am so terribly late for which I would like to apologize. I actually found the pictures interesting and helpful, and I would think they would make a good contribution to the supplement.

The text I found very well written, the figures clear and the conclusions are sound. I therefore recommend the manuscript for publication after minor revision.

Specific comments

Line 95 "This range...allow" – shouldn't this be "allows" (singular)?

Line 105: lowest, and those with both zooplanktivorous and piscivorous fish would have an intermediate level.

Could you elaborate on why you hypothesize that those have an intermediate level of pigmentation? I would assume that if zooplanktivorous fishes are present, they would prey on zooplankton anyway, no matter if there are piscivorous fishes in the same habitat. The only mechanisms I could think of (and you apparently do, too, as becomes apparent from sentences below) is the number of zooplanktivorous fishes being reduced but the fish abundance is not clearly presented in your manuscript (see line 118/119 – a little bit more detailed information already here in not only in the supplement (see line 123) would be appreciated by the reader, I believe).

Line 146: How did you divide the zooplankton populations?

Line 161: How many copepods did you study per replicate – is it five per replicate /container (see line 187) yielding 25 measurements or is it one per replicate. Please, clarify.

Legend Fig. 3: I believe it would be nice if you could add that this figure shows the effect of the removal of UV expose and the exposure to predatory clues. As it is, I had to go for and back between the results and the methods section to understand what exactly is presented.

Review form: Reviewer 2

Is the manuscript scientifically sound in its present form?

Yes

Are the interpretations and conclusions justified by the results?

Yes

Is the language acceptable?

Yes

Is it clear how to access all supporting data?

Yes

Do you have any ethical concerns with this paper?

No

Have you any concerns about statistical analyses in this paper?

Yes

Recommendation?

Accept with minor revision (please list in comments)

Comments to the Author(s)

Using a combination of population comparisons and a common garden laboratory experiment to test the combined effects of UVR and predation threats on the phenotypically plastic color morphs of low-latitude copepods. The manuscript is clearly presented and I have no serious concerns; the concerns I do have are intended to improve the clarity of the text.

Field sampling: some additional details are needed on the collection sites. Do they differ in area, depth, shade cover or other traits? For readers unfamiliar with the sites, some description would be useful. Likewise, only three sites were used (effectively, $n = 1$ per predation level). While not a critical flaw, some acknowledgement of this potential limitation is warranted.

Line 150: was 'fresh' water samples used for daily water changes? Presumably predator odours decay rapidly.

Review form: Reviewer 3

Is the manuscript scientifically sound in its present form?

Yes

Are the interpretations and conclusions justified by the results?

Yes

Is the language acceptable?

Yes

Is it clear how to access all supporting data?

Yes

Have you any concerns about statistical analyses in this paper?

No

Recommendation?

Accept with minor revision (please list in comments)

Comments to the Author(s)

In the presented study, the authors investigate the pigmentation of copepods in a natural “blue hole” system in the Bahamas. Most studies on zooplankton pigmentation are from higher latitudes which show highly variable levels of UV and predation stress throughout the year. Contrary to that few studies come from low latitude system where “threat” levels are assumed to be more constant. I agree with the authors that there is a lack of zooplankton studies focusing on the lower latitudes. The authors contribute to fill this gap with a study in a fascinating system of blue hole ecosystems. They combine field observations with an experiment testing for the influence of predator cues and UV radiation. The authors could confirm that in a habitat with predators, copepods had less pigmentation. However contrary to their expectation animals plastically lost pigmentation after the removal of UV.

I liked the study design, with its combination of field measurements and lab experiment. The study uses three blue holes of differing predation regimes and measures the pigmentation of the zooplankton in them. The field sampling and photo measuring and analysis seem to be have done properly.

The pattern follows the findings from higher latitudes with least pigments in the low latitudes. Given that there was only one pool per predation regime it would be good to discuss in one or two sentences about the generality of the field sampling results in the discussion part.

My comments are relatively minor and mostly focus on getting some additional information about the sites, and some more explanations to specific results.

I generally would love to get a bit more information about the blue holes, study site, maybe with a small map. Are the freshwater layers of the same depths? Can the copepods do vertical migration during daytime to escape both UV or fish predation?

Excerpt: at the end of the experimental period (10 day ...
(p.7, l.156)

Comment: I assume the sampling/measuring was done randomly with respect to the treatments over time. I was wondering whether the authors saw any effect of day time on pigmentation?

Excerpt: ...reducing stress to the copepods,
(p.7, l.152)

Comment: It is a bit unclear how this is reducing stress? By adding food?

Excerpt: floating atop marine ground water (31), represent a unique opportunity to investigate zooplankton pigmentation and
(p.4)

Comment: Are they always exposed to UV or can they escape to deeper water layers.

Excerpt: of a zooplanktivorous fish (Bahamas mosquitofish, *Gambusia hubbsi*), hereafter referred to as 'low-predation'; and Rainbow Blue Hole (24° 47' 6" N, 77° 51' 36" W) has no piscivorous fish and a high density of *G. hubbsi*, hereafter 'high-predation'. (p6, l 119)

Comment: Is it only fish predators in this system or could there be also other visual maybe invertebrate predators in the pools which would be invisible to the human observer.

Excerpt: Regardless of predation-risk treatment, the no- and low-predation risk populations lost pigmentation (in both redness and yellowness) in the absence of UVR, whereas the high-predation risk population increased pigmentation...
(p.9)

Comment: How much gene exchange is there between the populations? Could that somehow influence how adapted the populations are to the conditions in the different pools?

Excerpt: Figure 3. Change in mean pigmentation levels (Δa^* = redness, Δb^* = yellowness) of copepods ($n = 25$ per treatment X population) after exposure to the treatments in the laboratory experiment compared with the mean pigmentation of samples taken directly from the respective blue hole (hatched line). The values depicted here represent the least square means and the ± 2
(p.15,18)

Comment: I think it would be good to mention in the figure caption that this is the change due to lack of UV.

Comment: It would be great to plot the raw data maybe as a faint color behind the mean/SEs to get a feel of the variation in the population and to follow the call of Weissgerber et al 205 (DOI:10.1371/journal.pbio.1002128) to show more than means and SEs.

Decision letter (RSOS-190321.R0)

19-Jun-2019

Dear Mr Lee

On behalf of the Editors, I am pleased to inform you that your Manuscript RSOS-190321 entitled "Low-latitude zooplankton pigmentation plasticity in response to multiple threats" has been

accepted for publication in Royal Society Open Science subject to minor revision in accordance with the referee suggestions. Please find the referees' comments at the end of this email.

The reviewers and handling editors have recommended publication, but also suggest some minor revisions to your manuscript. Therefore, I invite you to respond to the comments and revise your manuscript.

- Ethics statement

- Data accessibility

If you wish to submit your supporting data or code to Dryad (<http://datadryad.org/>), or modify your current submission to dryad, please use the following link:
<http://datadryad.org/submit?journalID=RSOS&manu=RSOS-190321>

- Competing interests

- Authors' contributions

- Acknowledgements

- Funding statement

Because the schedule for publication is very tight, it is a condition of publication that you submit the revised version of your manuscript before 28-Jun-2019. Please note that the revision deadline will expire at 00.00am on this date. If you do not think you will be able to meet this date please let me know immediately.

Supplementary files will be published alongside the paper on the journal website and posted on the online figshare repository (<https://rs.figshare.com/>). The heading and legend provided for each supplementary file during the submission process will be used to create the figshare page,

so please ensure these are accurate and informative so that your files can be found in searches. Files on figshare will be made available approximately one week before the accompanying article so that the supplementary material can be attributed a unique DOI.

on behalf of Professor Emily Standen (Associate Editor) and Kevin Padian (Subject Editor)
openscience@royalsociety.org

Associate Editor Comments to Author (Professor Emily Standen):

Dear Dr. Lee,

Thank you for submitting your manuscript entitled Low-latitude zooplankton pigmentation plasticity in response to multiple threats. This is a clear and succinct paper that reports on the plastic effects both biotic and abiotic threats have on the pigmentation of zooplankton. I enjoyed reading your work.

Both reviewers have very minor comments and suggestions for clarifying your text in a few places. Please address these comments prior to re-submission.

Sincerely,
Emily Standen

Reviewer comments to Author:
Reviewer: 1

Comments to the Author(s)
Dear authors,

It was a pleasure to reading the manuscript on pigmentation plasticity in copepods from low-latitude habitats you have submitted to the Royal Society Open Access Journal. The Blue Holes

obviously provide an ideal opportunity to study community responses to environmental conditions, in this case UV exposure and predation pressure. It is, however, a little bit unfortunate that the community composition was so drastically different among the three holes, i.e. cyclopoids in no and high predation environment and calanoids in the intermediate system. This topic is addressed in the discussion (lines 240 ff) but I would suggest to removing this from results (maybe mention in the methods section), as you clearly state that such comparison does not yield much information.

In the detailed comments you will also find a question as to how many copepods were examined. I gather that it was always five individuals from each container, yielding 25 measurements for each group (see also Fig. 3 legend). It would be nice if you could state this more clearly in the methods section.

When I started working on the manuscript, I downloaded a zip file with the pictures of the copepods. Now, that I am finalizing my review, I tried to find the zip file on the web site again but I have not been successful, maybe because I am so terribly late for which I would like to apologize. I actually found the pictures interesting and helpful, and I would think they would make a good contribution to the supplement.

The text I found very well written, the figures clear and the conclusions are sound. I therefore recommend the manuscript for publication after minor revision.

Specific comments

Line 95 "This range...allow" – shouldn't this be "allows" (singular)?

Line 105: lowest, and those with both zooplanktivorous and piscivorous fish would have an intermediate level.

Could you elaborate on why you hypothesize that those have an intermediate level of pigmentation? I would assume that if zooplanktivorous fishes are present, they would prey on zooplankton anyway, no matter if there are piscivorous fishes in the same habitat. The only mechanisms I could think of (and you apparently do, too, as becomes apparent from sentences below) is the number of zooplanktivorous fishes being reduced but the fish abundance is not clearly presented in your manuscript (see line 118/119 – a little bit more detailed information already here in not only in the supplement (see line 123) would be appreciated by the reader, I believe).

Line 146: How did you divide the zooplankton populations?

Line 161: How many copepods did you study per replicate – is it five per replicate /container (see line 187) yielding 25 measurements or is it one per replicate. Please, clarify.

Legend Fig. 3: I believe it would be nice if you could add that this figure shows the effect of the removal of UV expose and the exposure to predatory clues. As it is, I had to go for and back between the results and the methods section to understand what exactly is presented.

Reviewer: 2

Comments to the Author(s)

Using a combination of population comparisons and a common garden laboratory experiment to test the combined effects of UVR and predation threats on the phenotypically plastic color morphs of low-latitude copepods. The manuscript is clearly presented and I have no serious concerns; the concerns I do have are intended to improve the clarity of the text.

Field sampling: some additional details are needed on the collection sites. Do they differ in area, depth, shade cover or other traits? For readers unfamiliar with the sites, some description would be useful. Likewise, only three sites were used (effectively, $n = 1$ per predation level). While not a critical flaw, some acknowledgement of this potential limitation is warranted.

Line 150: was 'fresh' water samples used for daily water changes? Presumably predator odours decay rapidly.

Reviewer: 3

Comments to the Author(s)

In the presented study, the authors investigate the pigmentation of copepods in a natural "blue hole" system in the Bahamas. Most studies on zooplankton pigmentation are from higher latitudes which show highly variable levels of UV and predation stress throughout the year. Contrary to that few studies come from low latitude system where "threat" levels are assumed to be more constant. I agree with the authors that there is a lack of zooplankton studies focusing on the lower latitudes. The authors contribute to fill this gap with a study in a fascinating system of blue hole ecosystems. They combine field observations with an experiment testing for the influence of predator cues and UV radiation. The authors could confirm that in a habitat with predators, copepods had less pigmentation. However contrary to their expectation animals plastically lost pigmentation after the removal of UV.

I liked the study design, with its combination of field measurements and lab experiment. The study uses three blue holes of differing predation regimes and measures the pigmentation of the zooplankton in them. The field sampling and photo measuring and analysis seem to be have done properly.

The pattern follows the findings from higher latitudes with least pigments in the low latitudes. Given that there was only one pool per predation regime it would be good to discuss in one or two sentences about the generality of the field sampling results in the discussion part.

My comments are relatively minor and mostly focus on getting some additional information about the sites, and some more explanations to specific results.

I generally would love to get a bit more information about the blue holes, study site, maybe with a small map. Are the freshwater layers of the same depths? Can the copepods do vertical migration during daytime to escape both UV or fish predation?

Excerpt: at the end of the experimental period (10 day ...
(p.7, l.156)

Comment: I assume the sampling/measuring was done randomly with respect to the treatments over time. I was wondering whether the authors saw any effect of day time on pigmentation?

Excerpt: ...reducing stress to the copepods,
(p.7, l.152)

Comment: It is a bit unclear how this is reducing stress? By adding food?

Excerpt: floating atop marine ground water (31), represent a unique opportunity to investigate zooplankton pigmentation and (p.4)

Comment: Are they always exposed to UV or can they escape to deeper water layers.

Excerpt: of a zooplanktivorous fish (Bahamas mosquitofish, *Gambusia hubbsi*), hereafter referred to as 'low-predation'; and Rainbow Blue Hole (24° 47' 6" N, 77° 51' 36" W) has no piscivorous fish and a high density of *G. hubbsi*, hereafter 'high-predation'. (p6, l 119)

Comment: Is it only fish predators in this system or could there be also other visual maybe invertebrate predators in the pools which would be invisible to the human observer.

Excerpt: Regardless of predation-risk treatment, the no- and low-predation risk populations lost pigmentation (in both redness and yellowness) in the absence of UVR, whereas the high-predation risk population increased pigmentation... (p.9)

Comment: How much gene exchange is there between the populations? Could that somehow influence how adapted the populations are to the conditions in the different pools?

Excerpt: Figure 3. Change in mean pigmentation levels (Δa^* = redness, Δb^* = yellowness) of copepods (n = 25 per treatment X population) after exposure to the treatments in the laboratory experiment compared with the mean pigmentation of samples taken directly from the respective blue hole (hatched line). The values depicted here represent the least square means and the ± 2 (p.15,18)

Comment: I think it would be good to mention in the figure caption that this is the change due to lack of UV.

Comment: It would be great to plot the raw data maybe as a faint color behind the mean/SEs to get a feel of the variation in the population and to follow the call of Weissgerber et al 205 (DOI:10.1371/journal.pbio.1002128) to show more than means and SEs.

Author's Response to Decision Letter for (RSOS-190321.R0)

See Appendix A.

Decision letter (RSOS-190321.R1)

25-Jun-2019

Dear Mr Lee:

On behalf of the Editors, I am pleased to inform you that your Manuscript RSOS-190321.R1 entitled "Low-latitude zooplankton pigmentation plasticity in response to multiple threats" has

been accepted for publication in Royal Society Open Science subject to minor revision in accordance with the editor's suggestions. Please find the editor's comments at the end of this email.

The Subject Editor has recommended publication, but also suggest some minor revisions to your manuscript. Therefore, I invite you to respond to the comments and revise your manuscript.

- Ethics statement

- Data accessibility

If you wish to submit your supporting data or code to Dryad (<http://datadryad.org/>), or modify your current submission to dryad, please use the following link:
<http://datadryad.org/submit?journalID=RSOS&manu=RSOS-190321.R1>

- Competing interests

- Authors' contributions

- Acknowledgements

- Funding statement

Because the schedule for publication is very tight, it is a condition of publication that you submit the revised version of your manuscript before 04-Jul-2019. Please note that the revision deadline will expire at 00.00am on this date. If you do not think you will be able to meet this date please let me know immediately.

on behalf of Professor Emily Standen (Associate Editor) and Kevin Padian (Subject Editor)
openscience@royalsociety.org

Subject Editor Comments to Authors:

Thanks for your attention to the reviewers' concerns. It has come to our attention that, while there is additional information about the field site, you did not publish the data or code to analyse it. Can you please rectify this in the final version? Thanks so much.

Author's Response to Decision Letter for (RSOS-190321.R1)

See Appendix B.

Decision letter (RSOS-190321.R2)

02-Jul-2019

Dear Mr Lee,

I am pleased to inform you that your manuscript entitled "Low-latitude zooplankton pigmentation plasticity in response to multiple threats" is now accepted for publication in Royal Society Open Science.

Kind regards,

on behalf of Professor Emily Standen (Associate Editor) and Kevin Padian (Subject Editor)
openscience@royalsociety.org

Follow Royal Society Publishing on Twitter: [@RSocPublishing](https://twitter.com/RSocPublishing)

Appendix A

Response to Referees' comments

Dear Professor Standen,

Thank you for your kind recommendation to publish in Royal Society Open Science. We are delighted! We would also like to take this opportunity to take all the reviewers for their time and effort in providing such constructive feedback. Please find a point by point list of the specific comments and reference to where we believe these concerns have been addressed. If there is anything further you require, please do not hesitate to contact.

Yours truly,
Marcus Lee

Reviewer 1

Line 95 "This range...allow" – shouldn't this be "allows" (singular)?

Addressed (p.5, l.109) *allow is correct in this instance as its not singular*

Line 105: lowest, and those with both zooplanktivorous and piscivorous fish would have an intermediate level.

Could you elaborate on why you hypothesize that those have an intermediate level of pigmentation?

Addressed (p.6, l.119) *We have explicitly stated the intermediate level is due to reduced predation pressure*

Line 146: How did you divide the zooplankton populations?

Addressed (p.8, l.164) *We have introduced a sentence to explain in more detail the setup of the treatments*

Line 161: How many copepods did you study per replicate – is it five per replicate /container (see line 187) yielding 25 measurements or is it one per replicate. Please, clarify.

Addressed (p.9, l.205) *We have made the sample size for each group more obvious in the statistical analysis section*

Legend Fig. 3: I believe it would be nice if you could add that this figure shows the effect of the removal of UV expose and the exposure to predatory clues. As it is, I had to go for and back between the results and the methods section to understand what exactly is presented.

Addressed (p.16, l.359) *We altered the figure legend as per request*

Reviewer 2

Field sampling: some additional details are needed on the collection sites. Do they differ in area, depth, shade cover or other traits?

Addressed (p.7, l.128) *We recognise the variation in blue hole features, but the full range of environmental traits surrounding these blue holes were not assessed here.*

Line 150: was 'fresh' water samples used for daily water changes? Presumably predator odours decay rapidly.

Addressed (p.8, l.171) *We discuss that we aimed to maximise the predator cues while preventing damage to the individuals and therefore performed water exchange every other day*

Only three sites were used (effectively, $n = 1$ per predation level). While not a critical flaw, some acknowledgement of this potential limitation is warranted.

Addressed (p.12, l.277) *We acknowledge the $n = 1$ of the pigmentation in natural populations*

Reviewer 3

It is a bit unclear how this is reducing stress? By adding food?

Addressed (p.8 l.166) *'reducing stress' By stress we intended to mean the mechanical disturbance of changing the water and the likelihood of injuring an individual through this process, as it would be possible to repeat this procedure more often yet risk mortality of focal individuals*

Given that there was only one pool per predation regime it would be good to discuss in one or two sentences about the generality of the field sampling results in the discussion part.

Addressed (p.12, l.277) *We mention that further samples would help increase the generality*

Are they always exposed to UV or can they escape to deeper water layers?

Addressed (p.12, l.298) *We make it more evident that diel vertical migration is possible in these blue holes, but currently there is no information as to whether they do conduct these movements in this system, so we refrain from discussing it too deeply*

I assume the sampling/measuring was done randomly with respect to the treatments over time. I was wondering whether the authors saw any effect of day time on pigmentation?

Addressed (p.8, l.177) *Yes, each container was randomly selected with respect to treatments and we did not note any difference in pigmentation changes on an hourly scale. We also did not look for that so it's possible there were minute changes not observable to the human eye.*

How much gene exchange is there between the populations? Could that somehow influence how adapted the populations are to the conditions in the different pools?

Response to comment about gene flow:

There has been no work conducted on dispersal across this system in zooplankton and as such hard to discuss with relation to this particular trait. It is possible that infrequent gene flow maintains the need for plasticity, and we do make reference to this in the discussion (p.13, l.330), but without specific evidence in this system it is hard to describe how adapted these organisms are to their native habitats.

Is it only fish predators in this system or could there be also other visual maybe invertebrate predators in the pools which would be invisible to the human observer?

Addressed (supplementary) *we highlight that predation risk is non-zero even for the 'no predation risk' environment, but due to the vast differences in fish versus invertebrate feeding rates we determine that this is likely not altering the outcome of the results reported*

I think it would be good to mention in the figure caption that this is the change due to lack of UV.

Addressed (p.16, l.359) *We altered the figure legend as per request*

It would be great to plot the raw data

Addressed (supplementary) *We completely agree that the variation in the population will be important however we believe this removes attention from the story of the figure. So for clarity we include the raw data in a separate figure in the supplementary information.*

Appendix B

Dear Professor Padian,

Thank you for accepting the manuscript for publication. In response to your comments, I believe you are referring to the raw data relating to turbidity measurements. I have now uploaded these data to dryad data repository and the code for the analysis performed. I hope this is satisfactory.

Yours truly,
Marcus Lee